



# Assessing synergistic radar and radiometer capability in retrieving ice cloud microphysics based on hybrid Bayesian algorithms

Yuli Liu[1], Gerald G. Mace[1]

[1]Department of Atmospheric Sciences, The University of Utah

*Correspondence to*: Yuli Liu (yuli.liu@utah.edu)

**Abstract.** The 2017 National Academy of Sciences Decadal Survey highlighted several high priority objectives to be pursued during the next decadal timeframe, and the next-generation Cloud Convection Precipitation (CCP) observing system is thereby contemplated. In this study, we investigate the capability of two CCP candidates, i.e. a W-band cloud radar and a submillimeter-wave radiometer, in ice cloud remote sensing by developing hybrid Bayesian algorithms for the active-only,

passive-only, and synergistic retrievals. The hybrid Bayesian algorithms combine the Bayesian MCI and optimization process to retrieve quantities and uncertainty estimates. The radar-only retrievals employ the optimal estimation methodology, while the radiometer-involved retrievals employ ensemble approaches to maximize the posterior probability density function. The a priori information is obtained from the Tropical Composition, Cloud and Climate Coupling (TC4) in situ data and CloudSat radar observations. Simulation experiments are conducted to evaluate the retrieval accuracies by

comparing the retrieved parameters with known values. The experiment results suggest that the radiometer measurements possess high sensitivity for large ice cloud particles, even though the brightness temperature measurements do not contain direct information on the vertical distributions of ice cloud microphysics. The radar-only retrieval demonstrates skill in retrieving ice water content profiles, but not in retrieving number concentration profiles. The synergistic information is demonstrated to be helpful in improving retrieval accuracies, especially in terms of ice water path. The end-to-end

simulation experiments also provide a framework that could be extended to the inclusion of other remote sensors to further assess the CCP observing system in future studies.

## 1 Introduction

The 2017 earth science decadal survey (Board, S.S and National Academies, 2019) identified five designated foundational observations to be pursued during the 2017-2027 time frame, and the Aerosols (A), and Clouds, Convection, and

Precipitation (CCP) are included as designated observables (DOs). In the preformulation study, the A and CCP DO's were merged to exploit synergies in the measurement systems. The objective of the preformulation study was to identify measurables that can achieve the science objectives of the DOs. As such, the study identified observing system architectures that maximize science benefit while limiting cost and risk. To narrow in on a set of viable architectures, the ACCP study relied on a suite of Observing System Simulation Experiments (OSSEs) aimed at addressing pixel-level retrieval

uncertainties and sampling trade-offs for various geophysical variables that were deemed important to achieving science goals.

The properties of ice clouds are among the critical geophysical variables in the CCP science objectives. Ice clouds play a significant role in modulating the energy budget of the earth system by absorbing upwelling long-wave radiation emitted from the lower troposphere and reflecting incoming solar short-wave radiation (Liou, 1986; Su et al., 2017). Studies suggest

that ice clouds are a net heat source to the climate system (Stephens and Webster, 1984; Ackerman et al., 1989; Berry and Mace, 2014) while contributing a positive feedback to the climate system (Zelinka and Hartmann, 2011).

The radiative effects of ice clouds depend on the vertically integrated and the vertical distribution of ice particle characteristics (Stephens and Webster 1984; Ackerman et al., 1988; Hartmann and Berry, 2017). The microwave radar and submillimeter-wave radiometry are two critical techniques for ice cloud remote sensing that are strongly synergistic when

combined (Buehler et al., 2012). The microwave radar reflectivity constrains ice cloud microphysical quantities in a vertically resolved sense while the submillimeter-wave radiometer constrains integrated mass and particle size. These two techniques are also highly complementary. The nadir looking microwave cloud radar provides high resolution of ice cloud vertical profiles but are limited to the along-track measurements, whereas the scanning submillimeter-wave radiometer has a wide swath but provides limited information about cloud vertical structure. Combing the strength of both observing sensors

enhances our capability to better acquire ice cloud spatial distributions.





Several retrieval algorithms have been developed specifically for ice cloud radiometry studies. All applicable algorithms that could be generally classified as statistical approaches and optimization approaches are under the framework of Bayes' theorem. The statistical approaches, including the Bayesian Monte Carlo Integration (MCI) (Evans et al., 2002; 2005) and the Neural Network (Jiménez, C et al., 2007; Brath et al., 2018), builds up an a priori database by randomly generating

atmospheric/cloud cases according to the a priori probability density function (PDF) and simulating instrument-specific measurements. The retrieval results are obtained through interpolation over the precalculated databases. To solve the sparsity of database cases in the measurement space, optimization algorithms are developed to maximize the posterior PDF. Evans et al., (2012) applied the Optimal Estimation Method (OEM) and Markov Chain Monte Carlo (MCMC) to retrieve ice cloud profiles from the Compact Scanning Submillimeter Imaging Radiometer (CoSSIR; Evans et al., 2005) observations during

the Tropical Composition, Cloud and Climate Coupling (TC4; Toon et al., 2010) experiment. Liu et al., (2018) proposed an ensemble methodology that does not use the gradient information but always relies on estimating posterior PDF to minimize the cost function. For the combined radar and radiometer retrievals, McFarlane et al., (2002) explored the synergistic concepts by retrieving liquid water content and effective radius profiles from millimeter wavelength radar reflectivity and dual-channel microwave brightness temperatures using the Bayesian MCI algorithm. Although this study worked on the

liquid cloud, the basic methodologies are applicable to the ice cloud remote sensing. Pfreundschuh et al., (2020) developed OEM algorithms for the upcoming Ice Cloud Imager radiometer (Kangas et al., 2014) and a conceptual W-band cloud radar to investigate to synergies between the active and passive observations.

The objective of this paper is to develop candidate algorithms for synergistic radar and radiometer retrievals to quantitatively assess the capability of sensing designated ice cloud geophysical variables for the next-generation ACCP observing system.

The algorithms for active-only, passive-only, and combined retrievals use a hybrid Bayesian framework, which combines the Bayesian MCI and optimization process to retrieve ice cloud quantities with uncertainty estimates. This paper is structured as follows: Section 2 provides an overview of the forward models and the simulated observations for testing the retrieval accuracies; Section 3 describes the hybrid Bayesian algorithms for the radar-only, radiometer-only, and synergistic retrievals in detail; Section 4 describes the a priori retrieval database using the statistics from in situ data and CloudSat radar

observations; Section 5 presents the retrieval simulation experiments and quantitative evaluations of the retrieval results; and

finally, Section 6 presents the summary and conclusions.

## 2 Simulated observations

### 2.1 Remote sensors

The remote sensors we evaluate in this study include a W-Band radar and a (sub)millimeter-wave radiometer both of which

are candidates in the ACCP observing system. The W-band cloud radar that we assume here is similar to the Cloud Profiling

Radar (CPR) in the CloudSat satellite (Stephens et al., 2008; Tanelli et al., 2008). The reflectivity measurement accuracy of

the target radar is 1.5 dB, and the minimum detectable reflectivity is -25 dB. The passive radiometer we consider is conical-

scanning with 16 horizontally polarized channels at the frequencies of $118 \pm 1.1$, $118 \pm 1.5$, $118 \pm 2$, $118 \pm 5$, $183 \pm 1$, $183 \pm$

2, $183 \pm 3$, $183 \pm 6$, 240, 310, $380 \pm 0.75$, $380 \pm 1.5$, $380 \pm 3$, $380 \pm 6$, 660, and 880 GHz. Most frequency channels are

centered on water vapor absorption lines. This radiometer has a 45° off-nadir angle and a 750 km swath width. Figure 1

shows the simulated clear-sky brightness temperature (BT) spectrum for a tropical atmospheric profile. All channels of the

submillimeter-wave radiometer considered in this study are positioned on the spectrum, and detailed views of the double

sidebands located on either side of the central frequency are also displayed.

### 2.2 Reference cloud scenes

We use simulations from the numerical Environment and Climate Change Canada (ECCC) model (Chen et al., 2018) run

using tropical atmospheric conditions to create the reference cloud scenes. The ECCC model outputs were made available to

the ACCP Science Impacts Team (Kollias, personal communication) and were originally created for use by the EarthCare

algorithm team (Illingworth et al., 2015). The major considerations in applying these atmosphere/cloud profiles are to assure

the independence between the ice cloud microphysics for reference and that in the retrieval database, but also to keep them

consistent in a geographic context. As will be discussed in section 4, the a priori database is created using in situ statistics

from NASA TC4 campaign that occurred in the Tropical Eastern Pacific. The ECCC model outputs water content and

number concentration profiles for several hydrometers including cloud ice, snow, liquid cloud, and rain. In this study,

however, we only use the frozen particle outputs, and we do not differentiate the cloud ice and snow but adding the water

content and number concentration of these two hydrometers to characterize the frozen particles. The reason for these

simplifications is still to be consistent with the a priori database that will be discussed in section 4. Currently, the retrieval

database we create does not contain liquid hydrometeors, and we do not distinguish between cloud ice and snow when

analysing the TC4 in situ data to capture the a priori statistics. All ECCC model outputs are interpolated according to the

CloudSat CPR range gate spacing that has 250-meter vertical resolution to mimic realistic remote sensing situations. A total

of 1280 atmosphere/cloud profiles along a latitudinal transect between -2.5° and 9° latitude are selected as the reference

cloud profiles for assessing the capability of the remote sensors.

**2.3 Radiative transfer model**

We develop the forward model for both active and passive simulations based on the Atmospheric Radiative Transfer

Simulator (ARTS; Buehler et al., 2005; Eriksson et al., 2011). ARTS is dedicated to radiative transfer calculations in the

millimeter and submillimeter spectral range. The recently published Single Scattering Database (SSD) for total random

orientation (Eriksson et al., 2018) and azimuthal random orientation (Brath et al., 2020) make ARTS a powerful tool for

investigating a variety of ice cloud properties. The ARTS forward model used in this study employs the two-moment scheme

which requires both ice water content (IWC) and number concentration (NC) to characterize the frozen particle size

distribution (PSD). The frozen particles are assumed to be randomly orientated, and their scattering properties are

represented by the "EvansSnow" habit from the ARTS SSD. Other configurations of the forward model remain the same as

the model used in Liu et al. 2018. It should be noted that the ARTS forward models used in simulating observations of the

reference cloud scenes are identical to the models used in the optimization retrieval algorithms, which means the systematic

biases from different particle habits or PSD schemes are not investigated in this study.

**2.4 Simulated observations**

Figure 2 shows the vertical distribution of IWC and NC for the selected reference cloud scenes along a latitudinal transect

and the corresponding W-band radar simulations. Compared to the number concentration, the radar reflectivity simulations

show a stronger tendency to follow the variations in IWC. Figure 3 shows the ice water path (IWP) and the corresponding

BT simulations for all ACCP radiometer channels. The correlations between the IWP changes and BT depressions are



evident. The channels with higher central frequencies are more sensitive to the change of water path, especially for small

changes in cloud ice on the order of 100 g m$^{-2}$. For the double sidebands with the same center frequency, the large

frequency-offset channels show higher brightness temperature values in clear sky conditions, and they have larger BT

depressions when encountering thick ice cloud layers. Figure 4 shows the scatterplot of the BT difference between

simulations in the clear-sky and cloudy conditions versus IWP for different channels. The 118 GHz channels demonstrate

sensitivity only when the IWP is over $10^3$ g m$^{-2}$. This is not surprising since the 118 GHz channels are primely designed for

sensing temperature profiles. For the 183 GHz and 380 GHz channels, the biggest BT differences are up to 50 K and 80 K,

respectively. Also, the 380 GHz channels simulations show more separation for the same IWP values, implying that the

high-frequency channels are more sensitive to the IWC vertical distributions. The BT differences for the 660 GHz and

880GHz window channels are noticeable even when the IWP is below 100 g m$^{-2}$, and the difference values could up to 110

K under our reference cloud scenes. These two channels make the ACCP radiometer capable of sensing thin clouds that are

usually composed of small particles. However, both 660 and 880 GHz show signs of saturation for IWP in excess of $10^3$ g m$^{-}$

$^2$.

**3 Hybrid Bayesian algorithms**

We developed different hybrid Bayesian algorithms for the radar-only, radiometer-only, and synergistic retrievals. All hybrid

algorithms combine Bayesian MCI with optimization processes to retrieve quantities and uncertainty estimates. Bayesian

MCI introduces prior information by generating an ensemble of atmospheric cases that are distributed according to the a

priori PDF, and it is highly efficient since the retrievals are done by interpolating the database cases and no more forward

model calculations are required. By assuming the uncertainties for different measurement variables to be independent, the

conditional PDF, which is also the posterior PDF, can be written as:

$$p_{cond}(x|y_{obs}) \propto exp\left(-\frac{1}{2}\chi^2\right) \qquad\qquad \chi^2 = \sum_{j=1}^{M}\frac{(y_{sim,j}-y_{obs,j})^2}{\sigma_j^2}, \qquad\qquad (1)$$

where $P_{cond}$ is the conditional probability of the measurement vector $y_{obs}$ given a particular atmospheric state $x$, $y_{sim}$ is the

simulated observation vector, and $\sigma_j$ is the uncertainty of observation and forward model for the $j$th channel. The retrieved





quantities and uncertainties are calculated by Monte Carlo Integration over the state vectors to find the mean vector and the associated standard deviation:

$$\bar{x} = \frac{\sum_i x_i exp\left(-\frac{1}{2}\chi_i^2\right)}{\sum_i exp\left(-\frac{1}{2}\chi_i^2\right)}, \qquad \sigma_{\bar{x}} = \sqrt{\frac{\sum_i (x_i - \bar{x})^2 \, exp\left(-\frac{1}{2}\chi_i^2\right)}{\sum_i exp\left(-\frac{1}{2}\chi_i^2\right)}} \qquad (2)$$

The biggest problem for the Bayesian MCI is the sparsity in the measurement space for a retrieval database with a finite

number of cases. If we increase the length of the observation vector or decrease the measurement uncertainties, the number of database cases matching the observation vector becomes smaller and the Bayesian MCI fails. When this happens, the optimization process is begun to maximize the posterior PDF.

**3.1 Radar-only retrievals**

The robust and efficient OEM method is employed as the optimization algorithm for radar-only retrievals. The fundamental

assumptions of the OEM algorithm are that the forward model is moderately nonlinear and that both prior PDF and conditional PDF are Gaussians. OEM maximizes the posterior PDF by minimizing the following cost function:

$$J = (F(x) - y)^T S_y^{-1}(F(x) - y) + (x - x_a)^T S_a^{-1}(x - x_a), \qquad (3)$$

where $F(x)$ is the forward model simulation, $S_y$ and $S_a$ are the covariance matrices for the measurement and prior uncertainties, respectively. In this study, the Levenberg-Marquardt minimization method (Rodgers, 2000) is implemented,

and the required Jacobian matrix is calculated by perturbing the ice cloud microphysical parameters (IWC and NC) in each pixel. The posterior error covariance matrix specified below is used to characterize the retrieval uncertainties:

$$S = (S_a^{-1} + K^T S_y^{-1} K)^{-1}, \qquad (4)$$

where $K$ is the Jacobian matrix to linearize the forward model. The covariance matrix $S$ is also derived based on the local Gaussian approximation and the forward model linearization assumption. The relative change of the cost function $J$ is

considered as the criteria for testing convergence. The OEM optimization terminates if the relative change of $J$ is below a specified threshold or the algorithm is over a certain number of iterations.





### 3.2 Radiometer-involved retrievals

The radiometer-involved retrievals that include the synergistic and radiometer-only retrievals do not employ the OEM algorithm in this study because the published OEM methods are not applicable under current testing circumstances. The

OEM algorithms involving BT measurements were developed in the following two studies. The first one was done by Evans et al., 2012, which computes the Jacobian matrix based on the adjoint modeling technique in the spherical harmonics discrete ordinate method for plane-parallel data assimilation (SHDOMPPDA) (Evans, K.F., 2007) radiative transfer model to make the evaluation of the gradient of cost function computationally feasible. The second one was developed by the ARTS community (Pfreundschuh et al., 2020), which calculate the BT difference sensitivity to the scaling parameters in a

normalized particle size distribution formalism proposed by Delanoe et al. (2005). This approach is not employed because a different PSD scheme is utilized in analysing the TC4 in situ data to capture the prior statistics, and a different prior Gaussian PDF characterized in terms of IWC and NC is used in this study. Besides, as pointed out in Pfreundschuh et al., (2020), the ARTS OEM method does not always satisfy the OEM fundamental assumptions requiring a nearly linear forward model, and the Jacobian evaluation is computationally very expensive. Based on the considerations above, we employ the

ensemble approaches instead to handle the radiometer-involved retrievals and defer the OEM analysis to future work. The ensemble approaches will be discussed in detail in the following two subsections.

### 3.2.1 Synergistic radar and radiometer retrievals

The synergistic radar and radiometer retrievals are done by extending the radar OEM algorithm to add the radiometer observations. The radar OEM algorithm provides the retrieved values as well as the associated uncertainty estimations

formulated in Eq. (4). Following this step, the Cholesky decomposition is implemented on the covariance matrix and an ensemble of random cases with a correlated Gaussian distribution around the radar retrieved vector is generated. This is done by decomposing the covariance matrix into a lower triangular form and then multiplying it by the standard normalized vectors. The corresponding BT simulations for the generated ice cloud profiles are subsequently computed using the ARTS radiative transfer model. After that, the ensemble cases are weighted according to their $\chi^2$ values that measure the distance

between the BT simulations and the input BT observation through Eq. (1), and the retrieval results and uncertainties are





computed by Monte Carlo Integration over the weighted ensemble cases to find the mean value and standard deviation, as indicated in Eq. (2).

### 3.2.2 Radiometer-only retrievals

We employ the Ensemble Probability Estimation (EnPE) algorithm as the optimization procedure for the radiometer-only

retrievals. The EnPE algorithm was first proposed by Liu et, al. (2018), and we continue to develop it as an optimization methodology. The EnPE algorithm has advantages in the following aspects. First, the algorithm does not rely on gradient information to move forward. Since the Jacobian calculations for BT observations are either complex to implement in the radiative transfer model or computationally expensive, the EnPE algorithm's characteristic of no Jacobian dependence makes it suitable for ice cloud profile retrievals that have high dimensional state vectors using advanced radiative transfer

models. Second, the EnPE algorithm is under the Bayesian MCI framework, which not only provides the theoretical basis but also offers a straightforward way to estimate the retrieval uncertainties associated with the retrieved quantities.

We describe the EnPE algorithm in a general way here, and more details could be found in Liu et, al. (2018). The EnPE algorithm stochastically explores the state vector space by sampling an explicit probability distribution function estimated from promising weighted cases found so far from the perspective of Bayesian MCI. The algorithm consists of two modules:

the estimation module numerically estimates the unknown continuous posterior PDF using the discrete cases with posterior values in the last ensemble, and the sampling module synthesizes new cases according to the accumulated PDF using the resampling approach and the covariance matrix.

Started from the situation where too few a priori database cases matching the observations, the estimation module artificially inflates the measurement uncertainties so that there are enough matches between the observation vector and the BT

simulations from the a priori profiles, and the posterior PDF is computed by:

$$P_{post} \propto P_{prior} * \exp\left(-\frac{1}{2\sigma_s^2}\chi^2\right), \tag{5}$$

where $P_{post}$ and $P_{prior}$ are the posterior PDF and prior PDF, respectively, and $\sigma_s^2$ is the inflation factor ensuring a minimum number of cases in one ensemble are within a specified $\chi^2$ threshold. $P_{prior}$ term is neglected in the first iteration since it is implicitly described by the distribution of the retrieval database cases. Following this step, the sampling module starts by



reselecting the cases according to their posterior value to multiply cases with high weights and kill cases with low weights, and the weights of the selected cases become equivalent again. The sampling module then adds correlated random noise to the selected cases using the two-point correlation statistics in the covariance matrix. The covariance matrix is computed using the posterior PDF based on Bayesian MCI:

$$Cov(m, n) = \sum_i (x_{i,m} - \overline{x_{i,m}})(x_{i,n} - \overline{x_{i,n}}) * P_{post,i} \tag{6}$$

The correlated noise generation step is essentially consistent with the Cholesky decomposition applied in the synergistic retrieval in section 3.2.1. However, since the covariance matrix here is not always positive definite, we use the empirical orthogonal functions (EOFs) to generate correlated random perturbations. The eigenvalues and eigenvectors of the covariance matrix in (6) are calculated, and the EOFs including 99.9% of the variance are used. The correlated Gaussian distributed elements are calculated by multiplying the Gaussian deviates by the square root of the eigenvalues to scale the

data based on the variance magnitude, and then multiplying them by the eigenvector to rotate back to the original axes. The correlated noise vectors are added to the selected cases in resampling step to further explore the state space. Once a new ensemble is synthesized and the corresponding BT simulations are computed, the algorithm evaluates these cases based on the prior PDF and likelihood PDF, and the optimization cycle starts again. As the iteration proceeds, the ensemble evolves and gradually becomes concentrated in the most likely area, compensating for the sparse distribution of the original retrieval

database. The iteration stops when meeting a specified criterion, and the remaining cases in the last ensemble are used to calculate the mean parameter values (retrieved values) and standard deviations (retrieved uncertainties) by Bayesian MCI.

Several updates regarding the EnPE algorithm are worth mentioning here. All updates are related to the precalculated retrieval database with the random cases distributed according to the a priori PDF. In Liu et al., (2018), the prior database is built up only relying on the numerical Global Environmental Multiscale (GEM; Côté et al., 1998) model outputs. The

disadvantages of this method are two-fold. First, the random cases cannot well represent the ice cloud distributions because there are many microphysical simplifications in such a numerical model that results in much less microphysical variability than exits in nature. Second, the reference cloud scenes come from the same GEM model, and the interference due to the close relations between these two datasets becomes inevitable since the datasets share the same GEM simulation parameters and initial conditions. In this study, we build up the retrieval database using the ice cloud microphysical statistics from in situ

measurements and spaceborne radar observations (again, see section 4.2 for more details). Accordingly, the random cases in

the updated retrieval database represent our prior knowledge of the atmospheric and cirrus clouds better, and they are also

completely independent from the reference cloud scenes for testing purposes. Further, since the random ice cloud profiles are

generated by statistically generalizing a relatively small number of cloud profiles that represent the prior information, a new

method is applied to deal with the regularization term ($P_{prior}$ in Eq. (5)) constraining the synthesized profiles to follow the

prior knowledge. This new approach captures more accurate a priori statistics, and it is applicable even when the a priori

PDF is highly non-Gaussian. This method will also be discussed in Section 4.2.

## 4 Prior information

The key element in implementing the Bayesian MCI is to build up the retrieval database, which generally consists of two

steps: creating random atmosphere and ice cloud properties that are distributed according to the prior PDF and computing the

simulated radar reflectivity or BT using the forward model. In this study, we separately develop two a priori retrieval

databases for radar and radiometer retrievals using the a priori statistics from TC4 in situ measurements and CloudSat CPR

observations.

### 4.1 Radar retrieval database

The realistic ice cloud microphysical probability distributions used for building up the radar retrieval database is obtained

from the in situ data from instruments flown in the TC4 campaign. The in situ ice particle size distributions are obtained

from the two-dimensional stereo (2D-S) probe and the precipitation imaging probe (PIP). The bimodal PSD scheme which

approximates both small and large particle distribution modes by gamma functions is used to fit the in situ data, and the ice

cloud parameters, including IWC, NC, and particle size are derived. More details about TC4 in situ analysis could be found

in Liu and Mace (2020). A multi-variant Gaussian distribution in temperature, *ln(IWC)*, and *ln(NC)* is used to capture the in

situ statistics, using the prior idea that the microphysical parameters are approximately lognormally distributed. Using a

multi-variant Gaussian function shows several advantages in generalizing the in situ statistics: first, it specifies the

microphysical PDF using a few parameters; second, it facilitates the radar OEM algorithm, which explicitly requires a

normally distributed prior PDF; third, it reasonably covers the space where the in situ probes fail to detect, which is

important since the random cases need to completely cover the possible parameter range. The parameters for the TC4 multi-

variant Gaussian function are summarized in Table 1. An ensemble of random cases (30,000 cases in this study) is sampled

from the Gaussian function, and the ARTS radar forward model is used to simulate the reflectivity for each random case.

Figure 5 shows the two-dimensional histogram for the microphysical quantities and reflectivity simulations in the radar

retrieval database. A fairly strong correlation between IWC and NC over the whole range is observed in the left panel. The

middle panel and the right panel indicate that the radar reflectivity simulations have a strong correlation with IWC in the

whole range, but its correlation with NC is much weaker.

### 4.2 Radiometer retrieval database

Apart from using the TC4 in situ microphysical statistics, we also use the CloudSat observations to acquire the critical

coherent vertical correlations to synthesize the random ice cloud profiles for creating the radiometer retrieval database. The

data we use include CloudSat radar reflectivity, CALIPSO lidar cloud fraction, and the corresponding ECMWF profiles of

temperature and relative humidity. The active remote sensing data profiles are combined with the TC4 cloud microphysical

probability distributions using the Bayesian MCI algorithm to create vertical profiles of microphysical properties that are

consistent with the measurements and the in situ statistics. After that, the cumulative distribution functions (CDFs) and EOFs

procedures are applied to capture the complete single-point and two-point statistics and then to create any number of

synthetic microphysical and thermodynamic profiles (100,000 profiles in this study) that are statistically consistent with the

Bayesian retrieval results. A comprehensive discussion on creating synthetic ice cloud profiles can also be found in Liu and

Mace, (2020).

As mentioned in section 3.2.2, we update the method for implementing the prior constraint that allows for using more

accurate prior statistics even when the a priori PDF is non-Gaussian. This method is consistent with the control vector

transformation concept applied in Evans et.al, (2012). The CDFs are used to capture the one-point statistics of different

parameters at different layers by sorting the variable from smallest to largest in value and calculating the sum of the assigned

equal probabilities up to each datum. The original ice cloud parameters are then represented by their percentile ranks, and the

correlations are also preserved in the rank matrix. Following that, the percentile rank matrix is transformed into a Gaussian

derivate matrix using the standard normal cumulative distribution function:



$$\xi_i = \Phi^{-1}(R(x_i)), \tag{7}$$

where $\Phi(\xi)$ is the standard normal cumulative distribution function, and $R(x_i)$ is the percentile ranks for different

parameters at different layers. For a new ensemble, the ice cloud profiles are transferred into Gaussian derivate matrices for

calculating the $\xi$ values, and the associated prior PDF values quantitating the strength of the prior constraints are directly

determined by the $\xi$ values. In this way, more realistic ice cloud statistics displayed in arbitrary functional forms are added

into the EnPE algorithm as the regularizations to make the algorithm more applicable.

Figure 6 shows the profiles of IWC, NC, temperature, and relative humidity for seven percentiles in the cumulative

distributions. Layers that are identified as clear are added with random Gaussian noise to prevent discontinuity in the CDFs.

The mean values for the added IWC and NC noise are $10^{-6}$ g m$^{-3}$ and 10 m$^{-3}$, respectively. The left two panels show that the a

priori IWC profiles cover the range from clear condition to about 10 g m$^{-3}$, and the NC profiles cover the range up to about

$10^6$ m$^{-3}$. The 50% curve only has meaningful values in the 11 to 13 km altitude range, indicating that the ice cloud particles

are mostly concentrated in this region. The 75% curve implying that a large majority of atmospheric conditions outside the 9

to 14km range are effectively clear. The right two panels show that the a priori temperature profiles have a small range of

temperature coverage under the tropical atmospheric conditions applied in this study, and the relative humidity profiles have

a large possible range, almost coving the entire possible values from 0 to 1.

The precalculated retrieval database provides a good opportunity for estimating the degrees of freedoms (DoF) for the ACCP

radiometer. The DoF describes the number of independent pieces of information in the radiometer measurement since some

channels provide redundant information. The DoF is usually calculated as the trace of the averaging kernel matrix based on

the Jacobian matrix (Rodgers, C.D., 2000), but a more general method described in Eriksson et al. (2020) is employed here

since the Jacobian matrix for BT is not available in this study. This method calculates the DoF in the measurement space

based on the EOF approach. The covariance matrix of a set of simulated BT is decomposed using EOF:

$$S_y = E \Lambda E^T, \tag{8}$$

where $E$ is the eigenvector and $\Lambda$ is the diagonal matrix containing the corresponding eigenvalues. The Gaussian

measurement noise in eigenspace is transformed back using the same eigen coordinate axes:

$$S_\Lambda = E S_\varepsilon E^T, \tag{9}$$

where $S_\varepsilon$ is the diagonal matrix that contains the square of measurement noise for different channels. The DoF is defined as

the number of diagonal elements in $S_y$ that are larger than the corresponding value in $S_A$ in the same place.

Figure 7 shows the DoF of the ACCP radiometer as the function of the ice water path (IWP) and integrated water vapor

(IWV). The necessary radiometer measurement noise is configured by mimicking to the CoSSIR uncertainty statistics that

are obtained from calibration target fluctuation statistics applied in Evans et al., 2012. The retrieval uncertainties for the

double-sideband channels including the 118 GHz, 183 GHz, and 380 GHz channels are set to 1.5K, 1.6K, and 2.3K,

respectively, and the uncertainties for the window channels including 240 GHz, 310 GHz, 660 GHz, and 880 GHz are set to

2.0K, 2.3K, 2.5K, and 4.0K, respectively. The DoF is computed only when the number of random cases in a certain IWV-

IWP range is larger than 10 to avoid noise interference. It can be seen that the DoFs increase with IWP. The DoFs are mostly

zero when the IWP values are smaller than 20 g m$^{-2}$, indicating the ACCP radiometer's limitation for IWP detection. The

DoFs generally equal to 1 in 20 to the 70 IWP range, and equal to 2 in the 70 to 110 IWP range. This analysis is consistent

with the plots in Figure 4, which show that only the 660Ghz and 880 GHz channels are sensitive to the thin cirrus clouds.

When IWP is over 300 g m$^{-2}$, the DoF is mostly between 6 to 8, and the DoF is over 10 very occasionally.

**5 Retrieval Simulation Experiment and Results**

In this section we present the analytical results for the radiometer-only, radar-only, and synergistic retrievals to assess the

capability of the objective ACCP remote sensors in retrieving ice cloud parameters. The retrieval experiments are performed

by inputting the simulated noisy radar reflectivity and BT observations into the hybrid Bayesian algorithms and then

comparing the retrieved parameters with the true values to determine the retrieval accuracy.

Some assumptions and configurations for the hybrid retrievals are summarized here. The independent Gaussian noise with a

standard deviation characterizing the radar measurement accuracy (1.5 dBz) is added to the simulated radar observations, but

we apply 4 dBz Gaussian noise in the Bayesian retrieval to also include the forward model uncertainty that is realized from

imperfect knowledge of ice crystal bulk density. The 4 dBz uncertainty is estimated based on the study of Mace and Benson,

(2017). Similarly, the Gaussian noise of 1K is added to the simulated BT observations in each channel to characterize the

measurement accuracy of the submillimeter-wave radiometer, but more realistic BT uncertainty estimations as specified in





section 4.2 are used in the hybrid Bayesian retrievals. For the radar-only and synergistic retrievals, the ice cloud layers with

noisy radar observations below the radar minimum sensitivity (-25 dBz) are ignored, and only the profiles of IWC and NC

are retrieved. Other atmospheric parameters, such as the profiles of water vapor, temperature, and pressure, are set to the true

values. For the radiometer-only retrievals, except for the IWC and NC profiles, we retrieve the water vapor profiles as well.

Some details in operating the hybrid algorithms are worth noting here. For the radar-only OEM retrievals, the initial state

vector is generated layer by layer based on the Bayesian MCI using the precalculated radar retrieval database discussed in

section 4.1. This process proceeds from top down, and the generated radar attenuation is used to correct the radar reflectivity

below. The a priori Gaussian PDF contains single-layer constraints with the Gaussian parameters listed in Table 1, but it

does not consider the vertical correlations between ice cloud microphysics at different layers. Besides, the radar OEM

retrievals always run in the logarithm space to help ensure that the assumed linearity is achieved. For the synergistic

retrievals, an ensemble of 500 cases is generated from the OEM retrieved uncertainties using the Cholesky decomposition

method. The following Bayesian MCI requires a minimum number of cases (25 cases here) matching the BT observation

within a specified $\chi2$ threshold. The $\chi2$ threshold is set as $M + 4\sqrt{M}$, where $M$ is the number of radiometer channels (Evans

et, al. 2005). If this criterion fails, we inflate the radiometer standard deviations in steps of a factor of $\sqrt{2}$ until reaching the

minimum number of cases, and the retrieval results and uncertainties are computed by MCI over the weighted cases, as

shown in Eq. (2). The Bayesian MCI computation is also done in logarithmic space. For the radiometer-only retrievals, the

EnPE algorithm generates 300 new cases in each iteration, and only 2 iterations at the maximum are permitted in this study

due to the computation limitation. The EnPE iteration stops when at least 25 cases within the $\chi2$ threshold are found in one

ensemble, or the number of iterations is over the limit. If there are not enough cases satisfying the $\chi2$ criterion in the lase

ensemble, we again inflate the BT measurement standard deviations until covering enough cases. The EnPE optimization

and the final MCI computations are done directly in the state space, not in the logarithmic space.

Figure 8 shows the direct comparison between the true values and the retrieval results for IWC and NC profiles along the

ECCC model transect. The results for the radar-only, radiometer-only, and combined retrievals are presented sequentially.

For the passive-only retrievals, the results suggest that there is essentially no information regarding the ice cloud vertical

profiles of both IWC and NC in the radiometer measurements. For the active-only retrievals, the retrieved IWC profiles

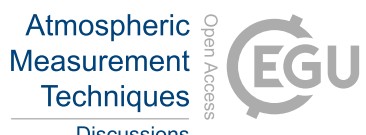

realistically reproduce the vertical structure of the reference cloud scenes. The retrieve values also correspond to the true

values in general, even though sometimes the retrievals tend to underestimate the IWC values, especially on the top of the

cloud ranging from 10 km to 15 km in height. By contrast, the active-only retrievals for NC profiles perform much worse.

The true NC values cover the range from 10 $m^{-3}$ to over $10^6$ $m^{-3}$, but the radar retrievals do not vary too much, usually

concentrating around domains in the $10^3$ $m^{-3}$ to $10^5$ $m^{-3}$ range. The retrieval results again illustrate that the radar

measurements are much more sensitive to the IWC variation compared to the NC variation. For the synergistic retrievals,

obvious perturbations can be observed for both IWC and NC profiles and the results become less smooth compared to the

radar-only retrievals. The added radiometer observations tend to correct the IWC underestimation discussed above.

Figure 9 shows the retrieved IWP values for the passive-only, radar-only, and combined retrievals based on the hybrid

Bayesian algorithms along the latitudinal transect. For the passive-only retrievals, the retrieval errors are comparable to the

active-involved retrievals over the entire range. The active-only retrievals show the tendency to overestimate the IWP for

thin clouds but underestimate the thick cloud IWP. The combined retrievals are developed from the radar OEM results, and

substantial improvements in IWP retrieval accuracy can be seen after adding the ACCP BT measurements.

Figure 10 shows the scatterplots of the retrieved parameters against the true values that are colored by density to further

visualize the retrieval performance. All statistical analysis for IWC below only applies to the grid points where the reference

IWC values are over $10^{-3}$ g $m^{-3}$. Similarly, the bottom limitations for NC and IWP analysis are 100 $m^{-3}$ and 10 g $m^{-2}$,

respectively. The scatterplots for IWC, NC, and IWP are shown in different columns, and the plots for passive-only, active-

only, and combined retrievals are shown in different rows. This figure could be directly compared to figures 7, 8, and 13 in

Pfreundschuh et al., (2020), and a similar phenomenon could be observed here. Starting from the IWC retrievals in the first

column, the passive-only retrievals show the largest deviations from the diagonal line, which is not surprising since the BT

measurements have low sensitivity to the vertical distribution of the ice cloud microphysics. The radar-only retrievals

provide much more accurate results. The scatter of points lies along the diagonal and the associated deviations are small. The

radar-only retrievals are observed to bias high for the tenuous cases and bias low when IWC values are high. The biases at

the high end are due to non-Rayleigh effects and attenuation. The prior constraint is another possible reason for causing both

low-end and high-end biases since the particles with extreme values possess small prior probability values. The combined



retrievals correct the high-end offset, and the scatter plots lie more along the diagonal. The rim of the scatter plots for the combined retrievals becomes less smooth, which is inevitable because the BT measurements are added through an ensemble

approach by generating random cases over a large possible range to statistically explore the state vector space. However, its systematic deviations are reduced compared to the radar-only retrievals, which is consistent with the analysis in Pfreundschuh et al., (2020). For the NC retrievals in the second column, the passive-only retrievals again show very little skill. The NC results from the radar-only retrievals do not follow the true values well. The retrieved values are always located in the range of $10^4$ m$^{-3}$ to $10^5$ m$^{-3}$ although the true values vary in a much wider range. The combined retrievals

improve the NC accuracies only when NC is over $10^4$ m$^{-3}$, but the overall performance is still poor. The IWP retrievals show very good performance overall. All retrieved values in different panels follow the true values with small associated deviations. The IWP results from passive-only tend to overestimate the true values when IWP is small and underestimate the true values when IWP is large. The underestimation performance could probably be corrected if more random atmospheric/cloud profiles covering the large IWP range are included in the precalculated radiometer retrieval database. The

active-only retrievals show a similar tendency, and significant improvements could be seen for the results from the combined retrievals.

Figure 11 shows the scatterplots of the absolute errors that are normalized by the retrieval uncertainties against the true values for different retrieval parameters. The normalized error is defined as

$$\delta_{error} = \frac{|x_{ret} - x_{true}|}{\sigma_{x_{ret}}},$$   (10)

where $x_{ret}$ and $x_{true}$ are the retrieved value and true value, and $\sigma_{x_{ret}}$ is the associated retrieval uncertainty. The $\delta_{error}$ measures how well the retrieved uncertainties reflect the actual retrieval errors, and it is another indicator for checking if the retrieval algorithms work well. The figure shows that the deviations of $\delta_{error}$ for radiometer-only retrievals are the largest, and the values spread from $10^{-2}$ to over $10^2$ for both IWC and NC. However, the areas with the highest density are always around the $\delta_{err} = 0$ line, and a large majority of cases could be found between $10^{-1}$ and $10^1$. By comparing the subplots for

the radar-only and combined retrievals, it is observed that the deviations of $\delta_{error}$ are increasing after implementing the ensemble approach to add BT information, but most cases still center around the $\delta_{error} = 0$ line within the $10^{-1}$ to 10 range, indicating that the absolute retrieval uncertainties are mostly within 1 order of magnitude of the actual retrieval errors. The



subplots suggest that the ensemble approaches applied to both radiometer-only and combined retrievals produce reasonable retrieval uncertainty estimations, which provides indirect evidence to support the good running of the hybrid retrieval

algorithms.

Figure 12 shows the scatter plots of the logarithmic error versus the true values for different parameters under different retrieval algorithms. The logarithmic error is defined as:

$$E_{log\,10} = log_{10}(\frac{x_{ret}}{x_{true}}), \tag{11}$$

The negative/positive values of $E_{log\,10}$ indicate that the retrieved values are smaller/larger than the true values, and the 0 dB

error representing the retrieved value and true value are identical. For the IWC retrievals in the first column, the radiometer-only retrievals show the strongest deviation, with the logarithmic errors spreading from -4 dB to +4 dB. However, the logarithmic errors tend to concentrate around zero when true IWC values are over $10^{-2}$ g m$^{-3}$, especially for cases around $10^{-1}$ g m$^{-3}$. The radar-only retrievals for IWC are more accurate, and the logarithmic errors are mostly between -1 dB and +1 dB. Still, the overestimations for the small IWC particles and the underestimations for the larger IWC particles are clear. The

combined retrievals help to improve the retrieval accuracies when IWC is large than $10^{-2}$ g m$^{-3}$. The combined retrievals, together with the radiometer-only retrievals shown on the top panel, suggest that the radiometer measurements possess high sensitivity for large particles with IWC over $10^{-2}$ g m$^{-3}$. For the NC retrievals in the second column, the log errors for the radiometer-only and radar-only retrievals both spread from -2 dB to +2 dB. The radiometer-only retrievals tend to have small log errors when true NC values are over $10^{-4}$ m$^{-3}$, but the radar-only retrievals do not exhibit skills in constraining NC over

the whole range. This phenomenon also agrees well with the findings in Pfreundschuh et al., (2020). The combined retrievals tend to improve the retrieval performance for particles with large NC values. Again, the combined and radiometer-only retrievals together suggest that the radiometer measurements are sensitive for particles with NC larger than $10^{-4}$ m$^{-3}$. For the IWP retrievals in the third column, the log error deviations are much smaller, mostly vary from -0.4 dB to +0.4 dB. The combined retrievals decrease the log errors over the entire possible IWP range.

Figure 13 displays the PDF of the logarithmic errors for different parameters under different retrieval methods and the corresponding CDF of the absolute logarithmic errors to summarize the logarithmic error distributions. As displayed in the left panels, the IWC logarithmic errors for radiometer-only retrievals cover a large range from -4 dB to 2 dB, and the radar-



only and combined retrievals are mostly concentrated between -1 dB to 1 dB. Compared to the error PDF for radar-only retrievals, the PDF for the synergistic retrievals has a smaller offset and smaller variance, even though the improvements are

not substantial. The NC retrievals displayed in the middle panels show little skill with the logarithmic error spreading from -2.5 dB to 2.5 dB.  As for the IWP retrieval displayed in the right panels, the passive-only and active-only retrievals show comparable errors, both distributing between -0.5 dB to 0.5 dB, and significant improvements for the synergistic retrievals is evident.

Figure 14 shows the quantitative values measuring logarithmic error distribution to compare the retrieval accuracy under

different retrievals. The top two panels show the mean values of the logarithmic errors and the associated IQR. The IQR is defined as the difference between the 75th percentile and 25th percentile. The mean and IQR values were also presented in Figure 11 in Pfreundschuh et al., (2020). However, since substantial differences in underlying assumptions exist in these two studies, the absolute values presented here could not be directly compared to those in Pfreundschuh et al., (2020). The differences are primarily reflected in the following aspects. The PSD schemes used in these two studies are not identical, and

the a priori PDF to constrain the optimization is significantly different. Also, the selection of the initial state vector to start the optimization differs. Further, as mentioned in section 2.3, we do not investigate the systematic biases coming from various particle habits, which results in much smaller absolute mean and IQR values compared to the results in Pfreundschuh et al., (2020). Nevertheless, the results could still be compared qualitatively to see if similar tendencies exist. For the IWC retrievals, the radiometer-only retrievals show the largest retrieval errors. Compared to the radar-only retrievals, the

combined retrievals correct the systematic biases, but the improvements in decreasing the IQR spreads are not evident. For the NC retrievals, the radar-only and radiometer-only results are both unsatisfactory and their IQR values are similar. For the IWP retrievals, the radiometer-only and radar-only show comparable capabilities, and the improvements from the combined retrievals are obvious since both biases and IQR deviations decrease. The tendencies observed in IWC and IWP retrievals here are generally consistent with the findings in Pfreundschuh et al., (2020). The bottom left panel shows the root-mean-

square deviation (RMSD) for different parameters to measure the deviations against zero. Not surprisingly, the radiometer-only retrievals have the highest number for both IWC and NC. The radar-only retrievals have a small RMSD value for IWC and a large RMSD value for NC, and the combined retrievals decrease the number on this basis.  Since the RMSD is easily

skewed by a few poor retrievals, the robust median errors that separate the higher half from the lower half in all the absolute

logarithmic errors are displayed in the bottom right panel. The median fractional bias is used to quantitatively assess the

relative improvements after adding BT measurements into the radar-only retrievals. The median bias for IWC retrievals

decreases from 0.34 to 0.28, indicating a 18% improvement, and the bias for NC decreases from 0.73 to 0.62, indicating a

12% improvement obtained from the BT information. The biggest improvement exists in IWP retrievals, which decreases the

median bias from 0.19 to 0.13, and the relative improvements reach 32%.

**6 Summary and conclusions**

In this study, we develop a suite of hybrid Bayesian retrieval algorithms for millimeter-wave radar and submillimeter-wave

radiometer to assess the ACCP observing system capability in sensing ice cloud microphysical quantities. Several new

algorithms are proposed here, and the algorithms could serve as alternative solutions for exploring the combined retrieval

concepts. The geophysical variables we investigate include the IWC, NC, and IWP. The hybrid Bayesian algorithms

combine the Bayesian MCI and optimization processes to compute retrieval quantities and associated uncertainties. The

radar-only retrievals employ the OEM optimization algorithm that uses gradient information to minimize the cost function.

The OEM is initialized by a state vector that is constructed by implementing Bayesian MCI to each reflectivity value in

different layers using the precalculated radar database. The necessary Jacobian matrix is calculated by perturbing the ice

cloud microphysical quantities in different layers. The radiometer-involved retrievals employ ensemble strategies to optimize

the ill-posed problem. The synergistic radar and radiometer retrievals are done by generating random cases from the radar

OEM results based on the Cholesky decomposition technique. The BT simulations are then computed, and the Bayesian

MCI is implemented to derive the final retrieval results. For the radiometer-only retrievals, the EnPE algorithm is applied to

statistically estimate the posterior pdf using the promising weighted cases. The estimation module and the sampling module

proceed iteratively to stochastically explore the state vector space. In addition, a new approach to implement prior constrain

that allow the a priori PDF to be highly non-Gaussian is proposed to make the ensemble algorithm more applicable.

We conducted simulation experiments to evaluate the accuracy of retrieving ice cloud quantities for different remote sensors.

The simulated noisy observations on a tropical transect of cloud profiles are input to the hybrid Bayesian algorithms, and the

retrieved parameters are compared to the known values to determine the retrieval accuracies. A tropical transect of cloud profiles that are simulated using the ECCC model is selected as the reference cloud scenes. This choice ensures the independence between the atmospheric/cloud profiles for testing and the vertical profiles in the a priori database. The main

conclusions from the presented results are summarized here:

1. The radiometer measurements do not have direct information about the IWC and NC vertical distribution. However, the BT measurements possess high sensitivity for large ice cloud particles with IWC values larger than $10^{-2}$ g m$^{-3}$ and NC values larger than $10^4$ m$^{-3}$.

2. The radar-only retrieval demonstrates skills in retrieving IWC profiles, but it literally does not exhibit capabilities in

retrieving NC vertical distribution. The radar-only retrievals for IWP have comparable accuracies to the radiometer-only retrievals.

3. The synergistic retrievals have evident improvements in retrieval accuracies compared to the radar-only retrievals. When using the median of the absolute fractional error as the quantitative parameter to evaluate the retrieval accuracies, the relative improvements after adding BT measurements for IWC, NC, and IWP are 18% and 12%, and 32%, respectively.

This paper provides an end-to-end idealized simulation experiment that sacrifices precise reality in order to demonstrate nuances in the various algorithms, and several disadvantages are worth mentioning. Firstly, the reference cloud scenes only contain frozen hydrometers, and the retrieval performance under more complex atmospheric scenarios is not investigated. Also, the forward model in this study only applies the EvansSnow particle habit, and the uncertainties caused by various particle habits are not considered. Secondly, the statistical characteristics are only derived based on selected

atmospheric/cloud profiles along a latitudinal transect. Since this subset with a finite number of profiles can hardly represent the realistic spatial distribution of ice cloud microphysics, the statistics we derive may differ from the characteristics of the entire possible atmospheric conditions. Thirdly, apart from the W-band radar and the submillimeter-wave radiometer, the ACCP observing system includes other remote sensors that would be highly helpful to improve retrieval accuracies for ice cloud remote sensing. For instance, highly accurate Doppler velocity measurements may allow for constraints on the ice

crystal bulk density that could significantly mitigate forward model uncertainties. The retrieval performance by combining other synergistic information content such as lidar remains to be investigated, and it will be explored in future work.

**Acknowledgments**

This research was supported by NASA Grants NNX15AK17G and 80NSSC19K1087 both administered from the Goddard Space Flight Center under the NASA ACE and ACCP programs, respectively. We thank for Prof. Pavlos Kollias at Stony Brook University for providing the atmospheric scenes from the ECCC model that we used for evaluating the synergistic retrievals. The TC4 in situ data for this study were collected by the members of the Stratton Park Engineering Corporation led by Paul Lawson, as well as Andrew Heymsfield at the National Center for Atmospheric Research. The CloudSat data were obtained from the CloudSat Data Processing Center at the Colorado State University's Cooperative Institute for Research in the Atmosphere (CIRA). All TC4 data are publicly and freely available in the NASA data archive at https://espoarchive.nasa.gov/archive/browse/tc4, and the CloudSat data are available at http://www.cloudsat.cira.colostate.edu/data-products.

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



Table 1. Ice particle microphysical statistics defining the a priori Gaussian probability distribution derived from the TC4 in situ data

|  | $\ln(IWC)$ $(g\ m^{-3})$ | $\ln(NC)$ $(m^{-3})$ | Temperature (K) |
|---|---|---|---|
| mean | -6.04 | 9.88 | 231.07 |
| std | 2.45 | 1.81 | 12.41 |
| correlation | $\rho_{\ln(iwc)-\ln(nc)} = 0.69$ | $\rho_{\ln(iwc)-tp} = 0.17$ | $\rho_{\ln(nc)-tp} = -0.10$ |




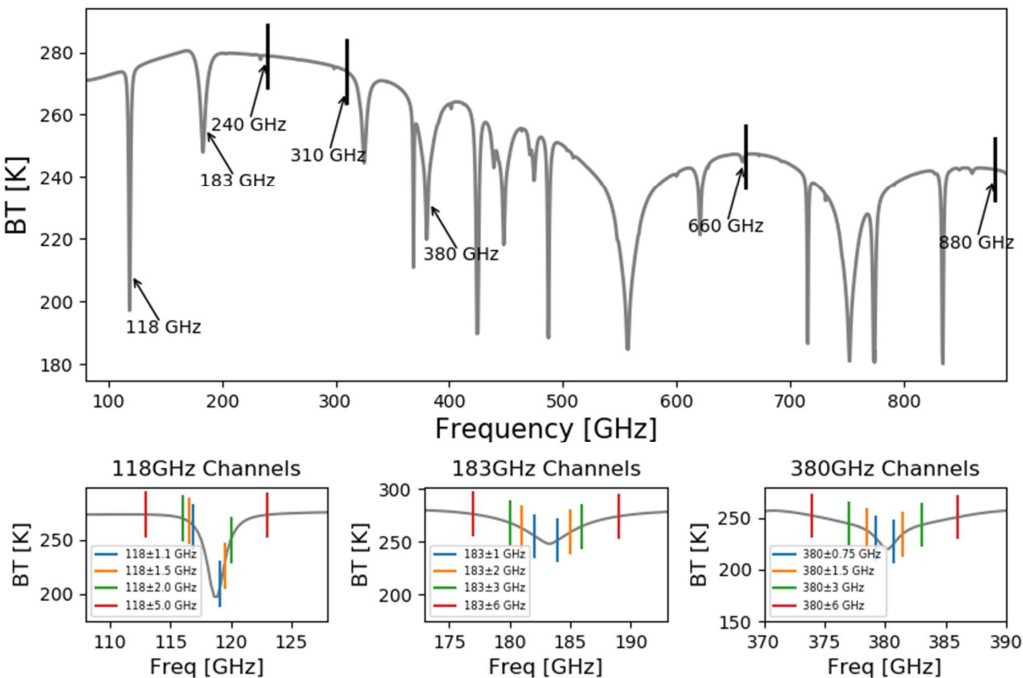

**Figure 1: Simulated clear-sky brightness temperature spectrum at a tropical atmospheric scenario. All ACCP radiometer channel positions and a detailed view of the double sidebands located on either side of a central frequency are present.**




**Figure 2: Vertical distribution of water content (WC) and number concentration (NC) for ice and snow particles along the selected latitudinal transact and the corresponding W-band radar reflectivity simulations. The radar simulations are computed using Atmospheric Radiative Transfer Simulator (ARTS) forward model.**




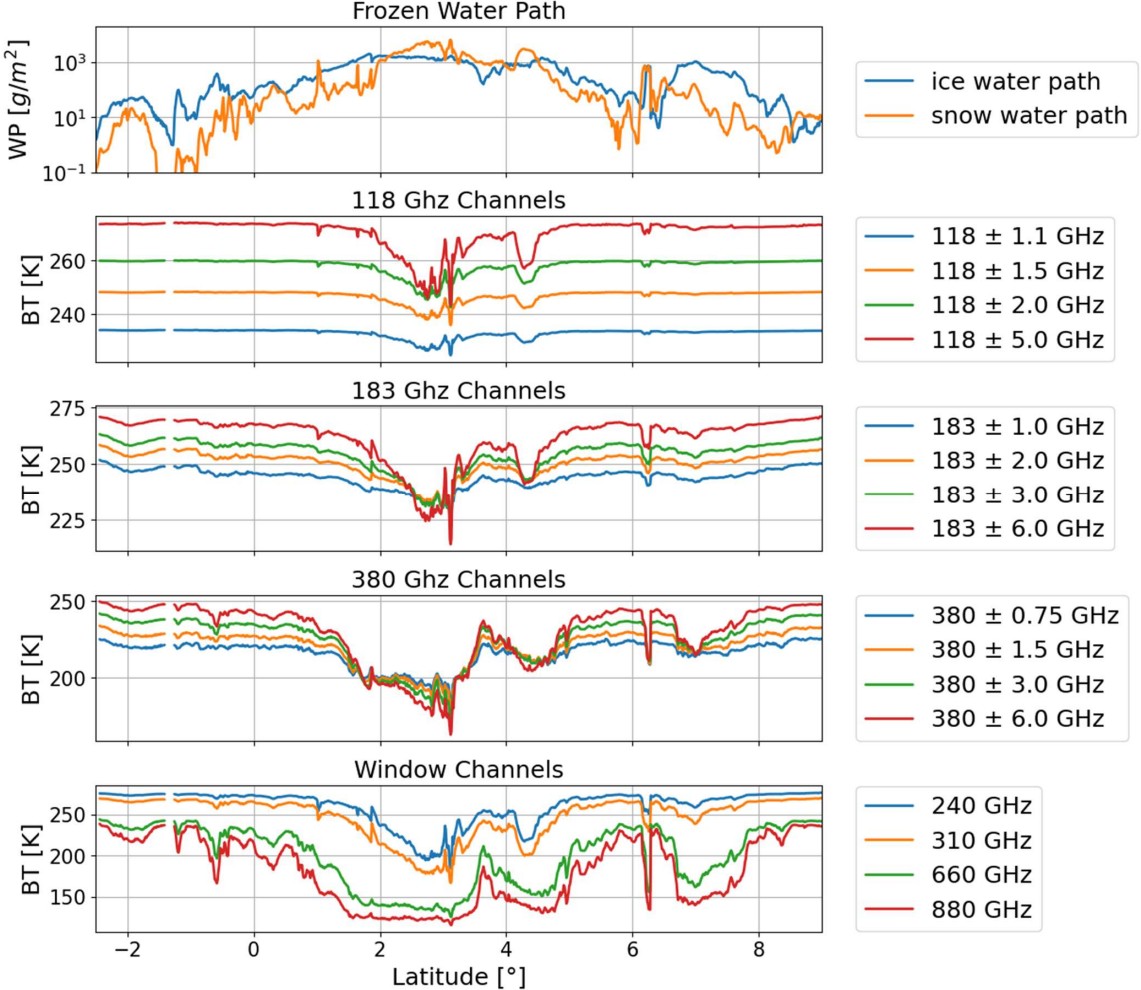

**Figure 3: Integrated water content for ice and snow particles for the selected latitudinal transect and the corresponding brightness temperature simulations for all ACCP radiometer channels.**






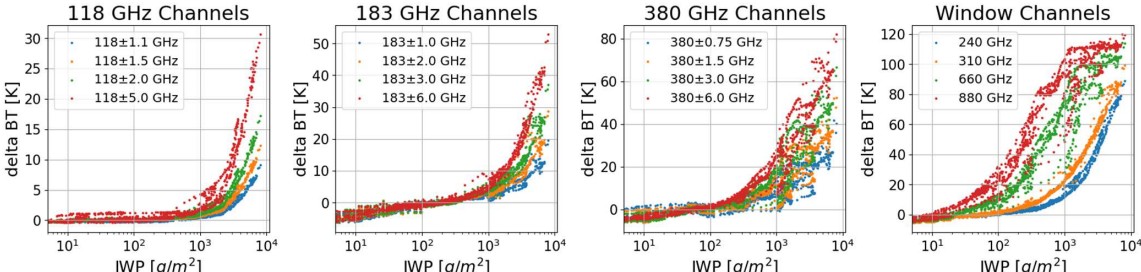

**Figure 4: Scatterplot of the brightness temperature difference between simulations in the clear sky and cloudy conditions as a function of ice water path for all ACCP radiometer channels.**

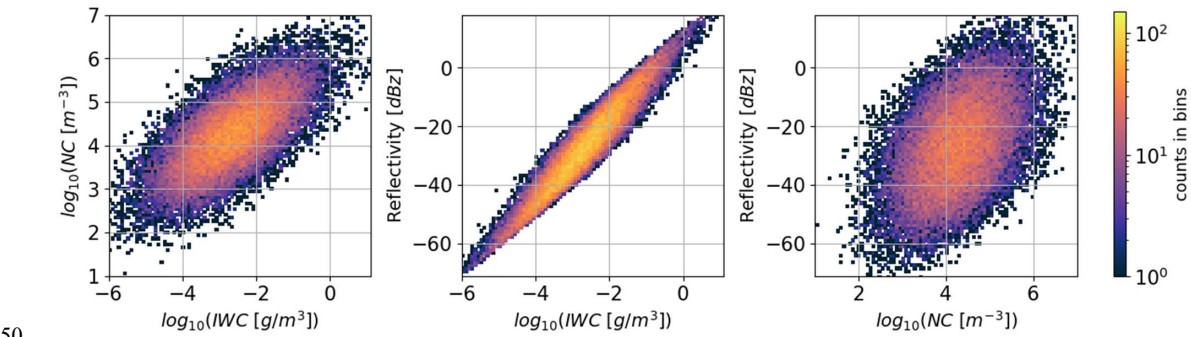


**Figure 5: Two-dimensional histogram for the microphysical quantities and the W-band radar reflectivity simulations derived from the random cases in the precalculated radar retrieval database.**








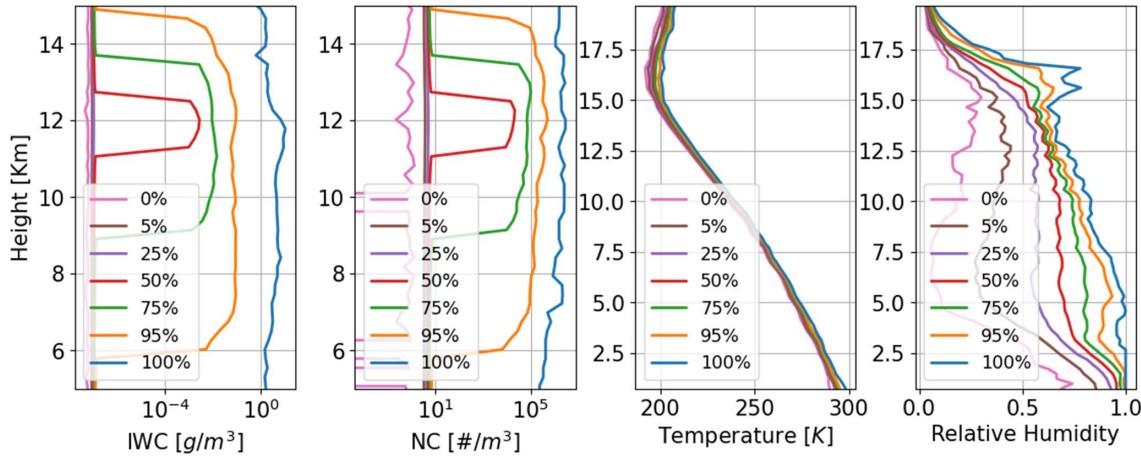

**Figure 6: Profiles of ice water content (IWC), number concentration (NC), temperature, and relative humidity for seven percentiles in the cumulative distributions for the random atmospheric/cloud profiles in the precalculated radiometer retrieval database.**





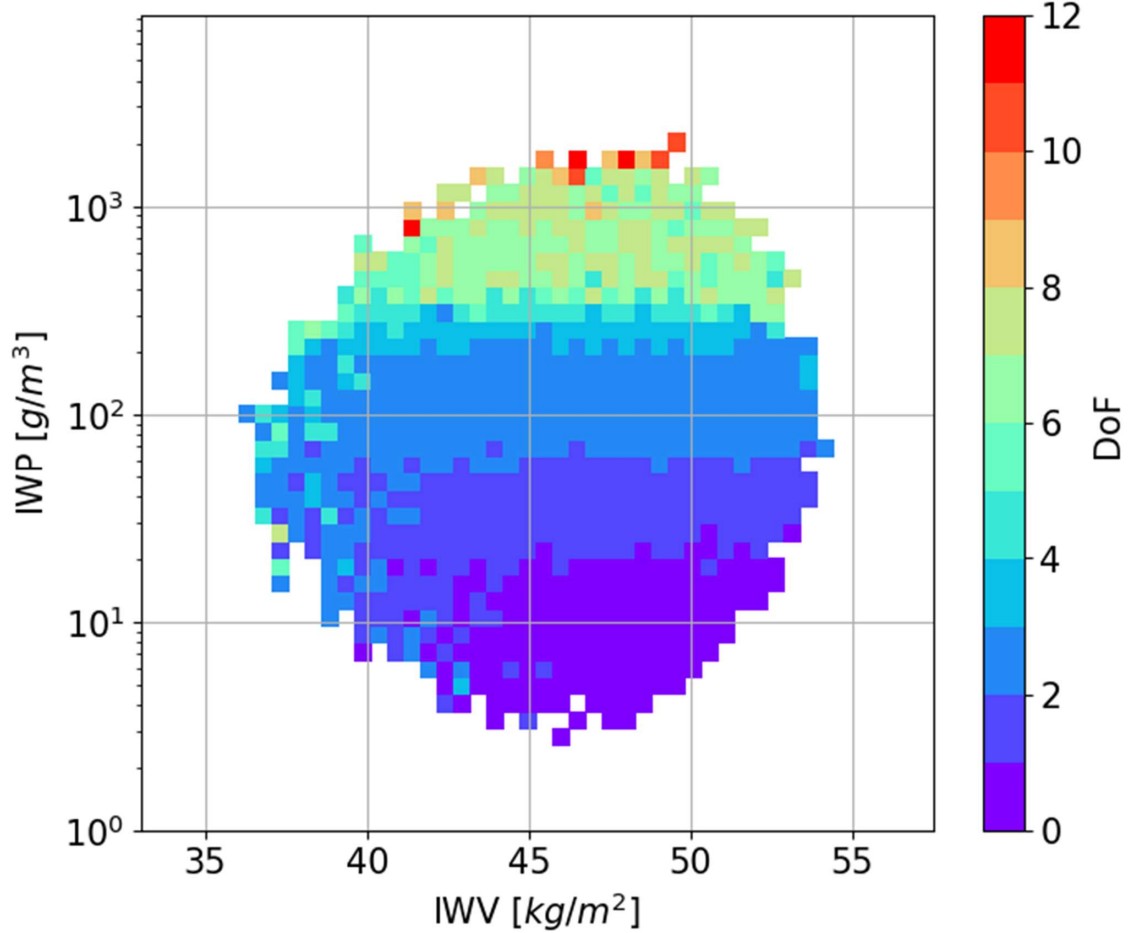


**Figure 7: The Degree of Freedoms (DoF) for the ACCP radiometer as the function of the ice water path and integrated water vapor. The DoF is estimated using the radiometer retrieval database that has 100,000 random atmospheric/cloud profiles.**



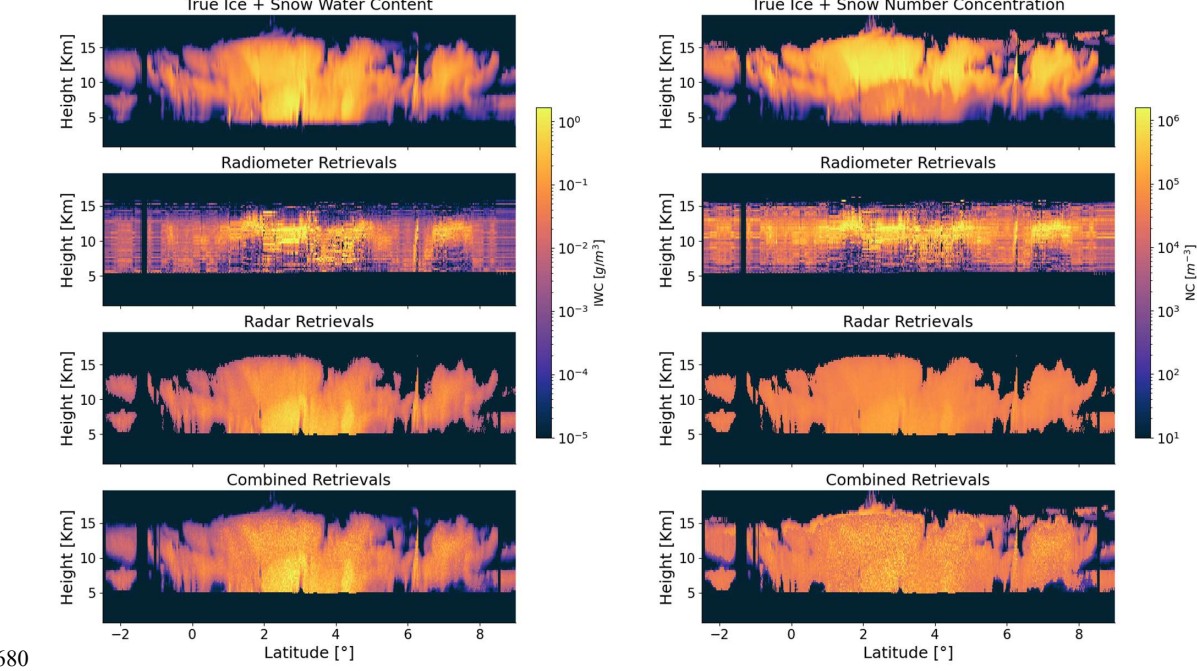


**Figure 8: Comparison between the true values and the retrieval results for ice water content and number concentration profiles along the selected latitude transect. The retrieval results for radar-only, radiometer-only, and combined are presented sequentially.**






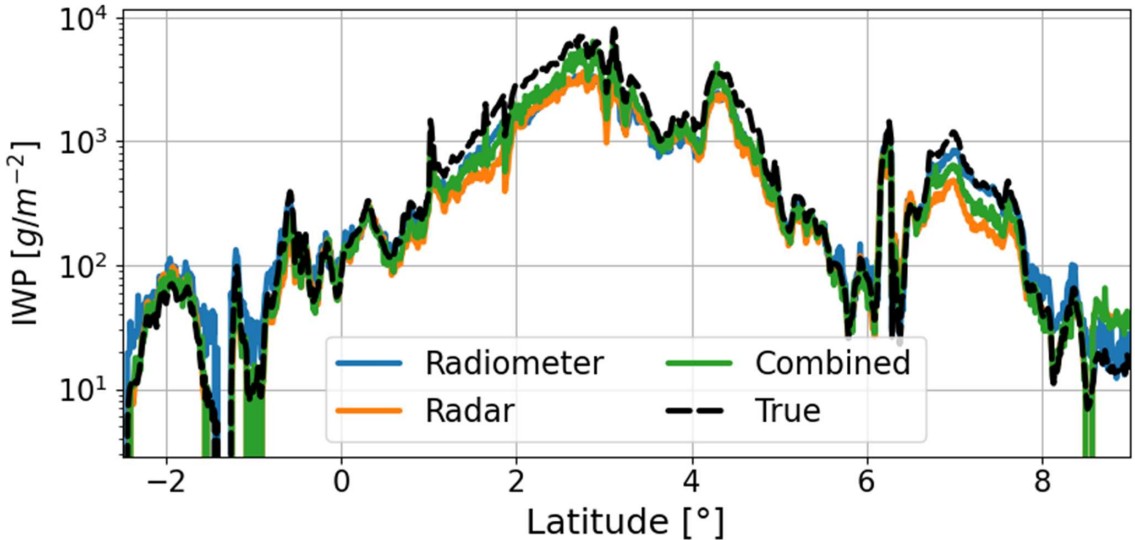

**Figure 9:** Direct comparison between the retrieved ice water path (IWP) and the true values along the latitudinal transect. The
passive-only, radar-only, and combined retrievals are all displayed.



**Figure 10: The scatterplots of the retrieved parameters against the true values that are colored by density. The scatterplots for ice water content (IWC), number concentration (NC), and ice water path (IWP) are shown in different columns, and the plots for passive-only, active-only, and combined retrievals are shown in different rows.**







**Figure 11: Scatterplots of the absolute errors that are normalized by the retrieval uncertainties against the true values. The normalized error is defined as:** $\delta_{error} = \frac{|x_{ret} - x_{true}|}{\sigma_{x_{ret}}}$, **where** $x_{ret}$ **and** $x_{true}$ **are the retrieved value and true value, and** $\sigma_{x_{ret}}$ **is the associated retrieval uncertainty.**



**Figure 12: Scatterplots of the logarithmic errors against the true values. The logarithmic errors is defined as:** $E_{log\,10} = log_{10}(\frac{x_{ret}}{x_{true}})$**, where** $x_{ret}$ **and** $x_{true}$ **are the retrieved value and true value.**




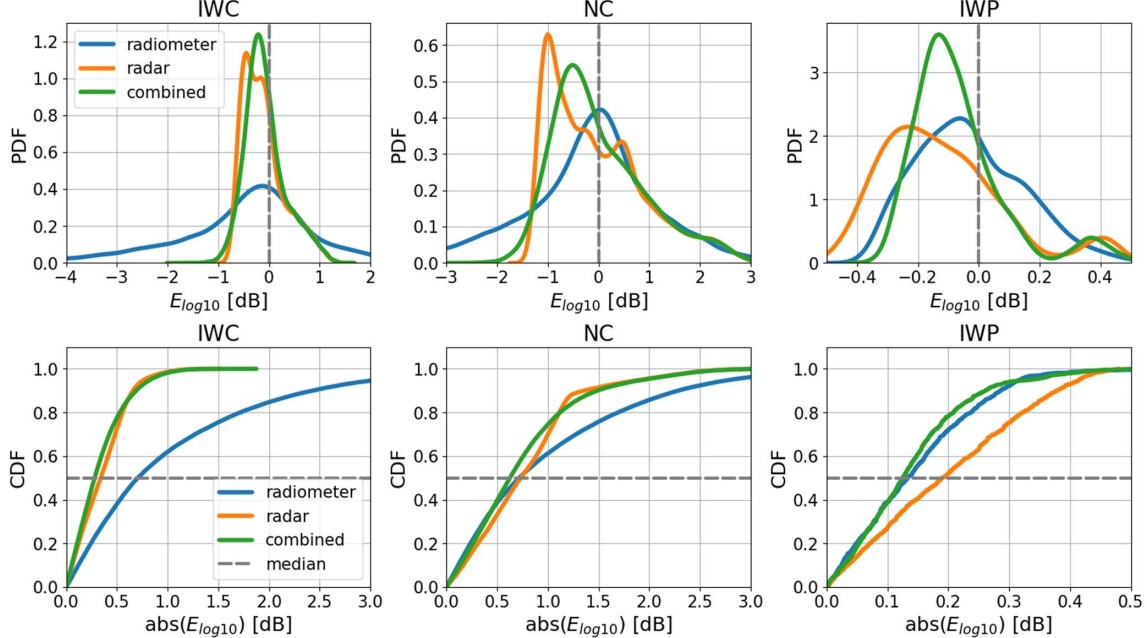

**Figure 13:** The top panels show the probability density function (PDF) of the logarithmic errors for different ice cloud parameters under different retrieval methods, and the bottom panels show the corresponding cumulative distribution function (CDF) of the absolute logarithmic errors.





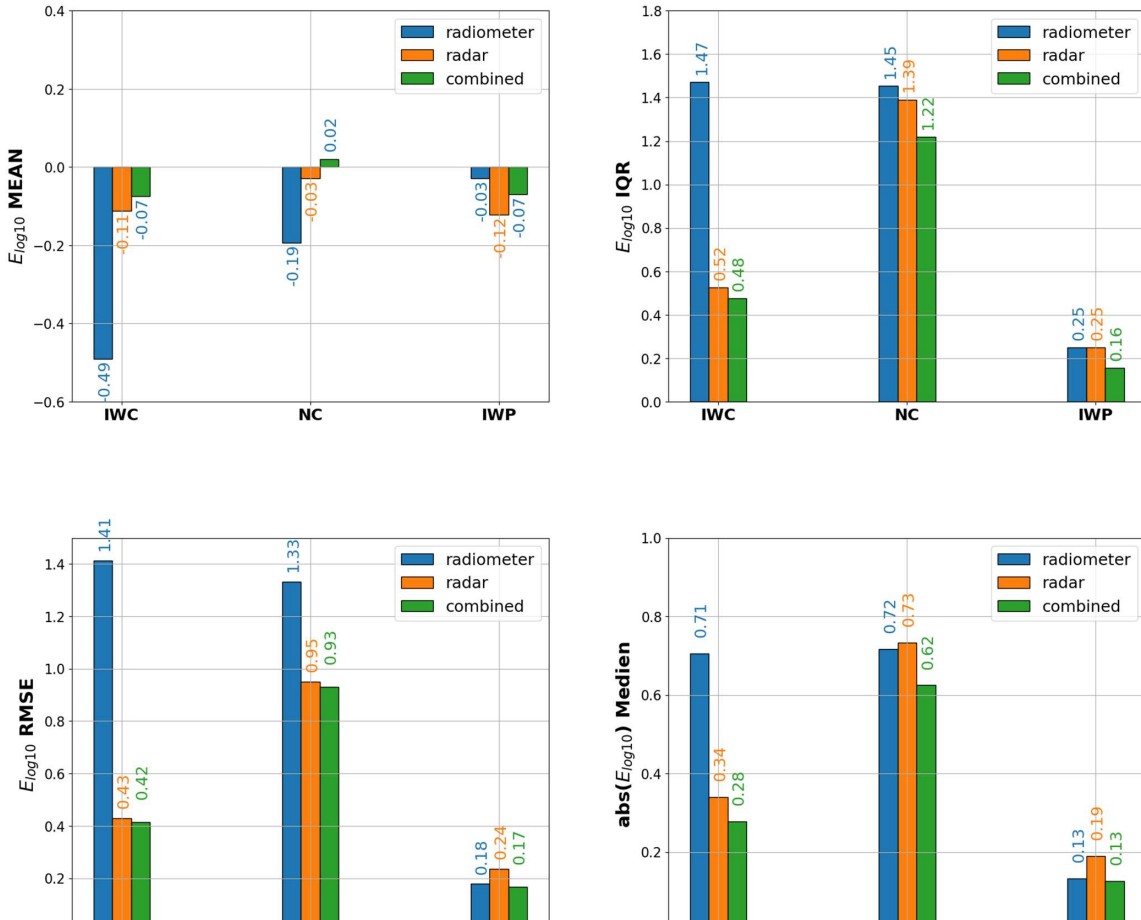

**Figure 14: The quantitative statistics of the logarithmic errors for the retrieved ice cloud quantities. The top panels show the mean values and the IQR, and the bottom left panel shows the root-mean-square deviation (RMSD) of the logarithmic errors. The bottom right panel shows the median values of the absolute logarithmic errors that separate the higher half from the lower half in all the retrieval error estimations.**