# Peer review of "Assessing synergistic radar and radiometer capability in retrieving ice cloud microphysics based on hybrid Bayesian algorithms"

_Atmospheric Measurement Techniques, 2021_

## Author Comment (AC1)

**Responses to Comments**

We sincerely thank Editor Eriksson, and two anonymous reviewers for the constructive and thoughtful comments.

Comments are in *blue italic lettering*, responses in black.

**Reviewer 1 Comments**

*General Comments:*

*The manuscript describes and evaluates a synergistic methodology to retrieve ice-cloud microphysics from synergistic radar and radiometer observations and provides. The evaluation is based on synthetic observations derived from a numerical weather prediction model. Overall, I find the manuscript well written and informative. However, the fact that radar and the radiometer are characterized by significantly different Field of Views (FOVs) should be addressed (or at least thoroughly discussed) in the manuscript. Specifically,*

*The radar considered in the manuscript is a nadir-looking instrument similar to the Cloud Profiling Radar (CPR) of the CloudSat mission, while the radiometer is a conically scanning instrument with a view angle of 450 from the nadir. While the radiometer's horizontal resolution is not specified, it is presumably coarser than that of the radar. It is not clear from the manuscript whether this aspect was considered in deriving the synergistic retrievals. In principle, one can account for the fact the that two instrument's FOVs are not the same, but the performance of the retrievals or the computational effort may be significantly different from those obtained when using simplifying assumptions. This should be discussed in the manuscript.*

Thanks for your general comments very much, and sorry for not mentioning the difference in horizontal coverage and viewing geometry between the active and passive sensors in the previous manuscript. This study focuses on investigating the synergistic retrieval performance when the fields of view are perfectly coincident. We have added specific statements in section 2.4 for the assumptions, as shown in Lines 113-120 in the revised manuscript. We also mention the simplifications in the summary section, as shown in Lines 507-510. The questions regarding the influence of footprints and viewing angles are great, and we will try to address them in future work once those characteristics are known.

*Minor Comments:*

*Page 5, Line 110. How exactly are the radiative transfer calculations done? Is the plane- parallel assumption made? Are any attempts to account for 3D effects made, such as slant-path calculations (Bauer et al., 1998)?*

The radiative transfer model in this study runs at the 1D atmosphere mode. As mentioned above, we assume that both sensors always have the same fields of viewing at nadir angle, and the 1D atmosphere could work well under such simplifications. The discussion could also be seen in Lines 113-120. 3D simulations are much more advanced in actual retrievals, but they are beyond the scope of the present study.

*Page 7, Line 155. I assume this means a finite difference scheme. If so, it is probably better to just call it a finite-difference scheme, as perturbation may be confused with the ensemble approach.*

This sentence has been rephrased to state the finite difference approach. The meaning of perturbations in the revised context should be clear now. Please check it in lines 160-162.

*Page 14, Line 323. These results are rather idealized than analytical.*

The present idealized retrieval experiments are done under a lot of assumptions such as the same fields of viewing, very simplified cloud species, single ice cloud habits, etc. We indicate these simplifications in different sections such as lines 109-111, lines 113-120, and the lines 507-510. We will keep improving the forward model and retrieval algorithms in the following work.

*Page 15, Line 335. Water vapor may be a significant source of uncertainties in the radar retrievals. It would be useful to investigate how the radiometer-retrieved water vapor impacts the synergistic retrievals.*

Thanks for your suggestion very much. Water vapors influence radiative transfer in (sub)millimeter significantly, and it is definitely worth investigating this problem more deeply. We currently only have IWC and NC as the state variables in the synergistic retrievals, as indicated in section 3.3 in lines 296-300. We mention that we neglect the impacts from water vapor variability in the summary section, as shown in line 525. A general way to see the impact of the water vapor in radiative transfer is through the figure 8, which shows the Degrees of freedom as a function of the integrated water vapor. More complicated simulation experiments will be done in the following studies to make the ACCP assessments more realistic.

---

## Author Comment (AC2)

**Responses to Comments**

We sincerely thank Editor Eriksson, and two anonymous reviewers for the constructive and thoughtful comments.

Comments are in *blue italic lettering*, responses in black.

**Reviewer 2 Comments**

*General comments:*

*The authors made a conceptual study about a synergistic microwave (MW) radar and MW/SubMM(submillimeter)-radiometer retrieval. The topic of the study is similar to the study of Pfreundschuh et al. (2020, AMT). Main differences are the different retrieval methods and that the authors' study is more restricted to a simpler atmospheric setup. The authors compare the synergistic retrieval to radar only and radiometer only retrieval. The synergistic retrieval is a combination of an optimal estimation (OEM) retrieval for the radar and Bayesian Monte Carlo integration for the radiometer. The basic story of the study is clear. The authors first define the hypothetical sensors, then explain the retrievals and retrieval databases, and show and discuss the results.*

*Despite the basically clear general story, the study is partially confusingly written and needs some restructuring. Some examples:*

*The retrievals are described in Sect. 3 but the actual values for the different thresholds, minimum numbers etc. are given in Sect. 5. This would make sense, if several retrieval configurations for each of the three retrievals were used, but this is not the case. The same holds for the actual retrieval quantities. They are given in the text of Sect. 5 but even there it is difficult to grasp what are the actual retrieval quantities, for example is log(IWC) or IWC a retrieval quantity.*

*In Sect. 4.2, it seems that the retrieval database is described but then the authors suddenly write about ensemble generation.*

*Update Sect. 3 explicitly stating for each retrieval the retrieval quantities, the measurement quantities, the assumed non-retrieval quantities, and all the retrieval specific a priori assumption, thresholds, minimum numbers etc.. Additionally, please do not mix ensemble generation and retrieval database.*

Thanks for the general comments regarding the manuscript structure. Following your advice, we add a new subsection 3.3 to explicitly state the measurement space and state space in this work, as shown in lines 287-295 in the revised manuscript. Also, the algorithm configurations such as the threshold and the number of cases in the ensemble approach are stated when we discuss the corresponding retrieval algorithms in section 3, such as the Lines 195-200, Lines 249-253 show. Further, the prior PDF calculation step has now been transferred to the EnPE algorithm description in section 3.2.2, as can be seen in lines 272-285.

*As I understand your paper, the idea is to combine a conical scanning radiometer with nadir pointing radar. This has two implications that you did not address in your study:*

*It is very unlikely that both sensors will have the same footprint.*

*Due to the different viewing geometry, both sensors have a different view on the atmosphere.*

*I think, it is not needed to expand your study to include these effects, because focusing on a 'best case' retrieval is still a formidable task. Nonetheless, you should explicitly mention that you neglect any footprint or viewing geometry effects and discuss at least briefly the implications.*

Thanks for your advice. We have added comments that we do not consider the footprint and viewing angles in this study in the simulated observation and the summary sections, as can be seen in lines 113-120 and lines 507-510. The influence of different fields of view and horizontal resolutions between active and passive remote sensors will be addressed in future work once we better understand the specifics of the sensors that will be flown.

*Section 1 Introduction*

*p. 3, l. 51: "The retrieval results are obtained through interpolation over the precalculated databases." Neural network retrievals are not an interpolation over a database.*

This sentence has been deleted.

*Section 2 Simulated observations*

*2.1 Remote sensors*
*Please include information about assumed footprint sizes and write explicitly the assumed viewing geometry of both sensors.*

The footprint and scanning modes of the radar have been added, as shown in lines 69-70. The assumption on the viewing geometry when simulating the observations and conducting retrievals are specified in section 2.4, as shown in lines 116-120.

*p. 4, l. 77-78: Put the channel description including noise, channel number, main spectral feature (H2O-line, O2-line, window…), etc. into a table and refer to it.*

*p. 4, l. 80: Please insert a sketch with the viewing geometry of the sensors.*

The noise characteristic of the candidate radiometer is listed in Table 1. However, we are not allowed to disclose too many details beyond the critical parameters in the simulation experiments because it will harm the competitive postures of the instrument teams. Also, the final instruments parameters may differ from what was assumed in the study. More information about the instrument designs will be published by the ACCP instrument teams when those instruments are established and known, as we indicate in lines 77-79.

*p. 4, l. 79: "Most frequency channels are centered on water vapor absorption lines." Change to something like this: 'The 183 GHz and 380 GHz channels are centered around H2O lines and the other channels are centered around the O2-line or a place within the window region.'*

This sentence has been revised following your advice, as shown in Lines 73-75.

*p. 5, l. 94-95: "The reason for these simplifications is still to be consistent with the a priori database that will be discussed in section 4." Did you consider the possible errors due to this simplification?*

We do not investigate the consequences of this simplification in this work, but we do continue to improve the simulation experiments including the differentiation of hydrometers to better assess the ACCP remote sensor. More result discussions in this aspect will be done in future work.

*p. 5, l. 97-100: What is the horizontal spacing between each model profile and what is the actual horizontal grid size of the ECCC model?*

The horizontal resolutions of the ECCC model and the dataset we used are both 250 meter, as shown in line 96.

*2.3 Radiative transfer model*

*Does the radar simulator includes attenuation?*

Yes, the radar forward model includes the attenuations.

*p. 5, l. 106-108: Please add what kind of particle size distribution (PSD) including used parameter and constants you used.*

The particle size distribution used in the ARTS model has been added, as shown in lines 113-114.

*p. 5, l. 109-110: Please include briefly the other forward configurations here and not by just referring to the other paper.*

More information about the ARTS model, including the scattering solvers, gas absorption database, and the surface module has been added, as shown in lines 101-103.

*2.4 Simulated observations*

*p. 5-6, l. 116-119. In the text you wrote cloud ice and in your figure 3, you distinguish between snow and ice but in Sect. 2.2 (p.4, l. 93), you write: "...we do not differentiate the cloud ice and snow...". Please be consistent or explicitly state in the corresponding lines you distinguish between ice and snow. Otherwise it can be confusing.*

The plots in figure 3 have been updated.

*3 Hybrid Bayesian algorithms*

*p. 6, l. 135: "...and it[BMCI] is highly efficient since the retrievals are done by interpolating the database cases" BMCI is strictly speaking not an interpolation.*

This sentence has been corrected, as can be seen in line 142.

*3.1 Radar-only retrievals*

*p. 7, l. 158: "…where K is the Jacobian matrix to linearize the forward model." Is K the Jacobian matrix of the retrieved state or of the a priori state? Please clarify within the text.*

The K is the Jacobian matrix of the retrieved state, as shown in line 165.

*3.2.1 Synergistic radar and radiometer retrievals*

*p. 8, l. 182-183: What are "standard normalized vectors". Please explain it within the text. Furthermore, do you generate only profiles of IWC and NC or do you generate also temperature and humidity profiles?*

The "standard normalized vectors" has been changed to "a vector of standard Gaussian deviates", as shown in line 189. The state variables in different algorithms have been summarized in section 3.3 in lines 296-300. Since the ensemble is generated from the covariance matrix obtained by the radar OEM algorithm, the state vectors in synergistic and in radar-only retrievals are consistent, and they only contain IWC and NC.

*3.2.2 Radiometer-only retrievals*

*p. 9, l. 197-202: I understand that you want to give a brief explanation of your retrieval. Nonetheless, your algorithm is complex. Therefore, I suggest to add a flowchart/algorithm chart especially in the view that you want to give a brief explanation of your retrieval algorithm.*

It is a good idea to add a flowchart to summarize the improvements of the EnPE algorithm and to make the algorithm more understandable. This flowchart has been added in Figure 5.

*p. 9-10, l. 209-211: "Following this step, the sampling module starts by reselecting the cases according to their posterior value to multiply cases with high weights and kill cases with low weights, and the weights of the selected cases become equivalent again." Please rephrase it. The sentence is hardly understandable.*

This sentence has been corrected, as shown in lines 227-228.

*p. 10, l. 211-214: "The sampling module then adds correlated random noise to the selected cases using the two-point correlation statistics in the covariance matrix. The covariance matrix is computed using the posterior PDF based on Bayesian MCI…" I do not fully understand what do you mean with "cases". Is this an atmospheric state or is this the whole database entry including brightness temperature and full atmospheric state? Is the mentioned covariance matrix calculated from the full atmospheric state or just from the retrieval quantities? Please clarify within the text.*

The "cases" represent the selected state vectors from the resampling module, and we mention it in lines 242-244. The covariance matrix is only computed for the retrieval quantities, as shown in line 230.

*p. 10, l. 218-221: Please consider to use mathematical formulas and equations.*

Formula of this step has been added, as shown in Eq. (8)

*p. 10, l. 222-223: "…the algorithm evaluates these cases based on the prior PDF and likelihood PDF,…"
Please explain, what "likelihood PDF" and "prior PDF" in that context mean? Is "prior PDF" the PDF of
the previous iteration? Please clarify within the text.*

The "likelihood PDF" is calculated in the measurement space, and we have changed it to
"conditional PDF" to be consistent, as seen in line 246. The prior PDF is discussed in the last part of
subsection 3.2.2, as shown in lines 272-285. Also, clear mathematical relationships could be found
in the flowchart shown in Figure 5.

*Section 4 Prior information*

*I would suggest to rename this section to 'retrieval databases', because this is the actual topic of this
section.*

The section has been renamed as retrieval database.

*4.1 Radar retrieval database*

*Please add a sentence for what you need this database, because at first view it seems strange to have a
retrieval database for an OEM retrieval.*

The discussions have been added in lines 321-326.

*p. 11, l. 254: How and where do you get the temperature information?*

The temperature is measured by the Meteorological Measurement System on the DC8 aircraft
platform, as shown in lines 309-310.

*4.2 Radiometer retrieval database*

*p. 13, l. 305-306: According to Eriksson et al. (2020, AMT) and the citation therein "E" is a matrix with
each column an eigenvector. Please correct.*

The meaning of E has been corrected, as shown in line 356.

*p. 14, l. 313-316: Please add the noise quantities to a table with the other channel specific properties,
see also my comment about 2.1 Remote sensors.*

The noise quantities have been added in Table 1, and we specify the measurement uncertainties
used in the retrieval experiments in lines 293-295.

*Section 5 Retrieval simulation experiment and results*

*p. 14, l. 331: "Similarly, the Gaussian noise of 1K is added to the simulated BT observations in each
channel to characterize the measurement accuracy of the submillimeter-wave radiometer,…" Why do
you add only 1 K of noise to the simulated observations? The noise values in given in Sect. 4.2 are much
higher. This seems to make no sense.*

The noise added into the simulated observations and that used in the retrievals are now consistent, as stated in section 3.3 in lines 293-295.

*p. 15, l. 336: "For the radiometer-only retrievals, except for the IWC and NC profiles, we retrieve the water vapor profiles as well." The sentence is confusing. Please rephrase.*

This sentence has been deleted, and the state variables for all retrievals are now specified in section 3.3 in lines 296-300.

*p. 15, l. 352-353: "The EnPE optimization and the final MCI computations are done directly in the state space, not in the logarithmic space." Some lines above (l. 348) you wrote "The Bayesian MCI computation is also done in logarithmic space". Please explain why do make a difference?*

We find that the EnPE passive-only algorithm works better in the non-log state space, as we indicate in lines 299-300.

*p. 15, l. 356-357: Why is the radiometer retrieval (Figure 8) so noisy?*

*p. 16, l. 363-365: Do you have any idea, why the synergistic retrieval is noisier than the radar-only retrieval?*

The ensemble approach generally makes the results noisier, but I am not sure if the noise is in a reasonable range. It seems like the retrievals are especially noise for the thin cloud with IWP smaller than 100 g m-2. We will try other methods to see we can make it more smooth in the following work.

*p. 16, l. 380-381: What do you mean with "non-Rayleigh effects and attenuation"? Please explain within the text.*

This description has been deleted.

*p. 16-17, l. 371-396: Please add some discussion about the consequences that you use a combined PSD for snow and ice within your retrieval. For me, it seems, that some of the bias in IWC of the radar retrievals is due to fact that you do not separate between ice and snow.*

We add a sentence saying a possible reason for radar-only bias is that we do not differentiate the cloud ice and snow in the forward model, as shown in line 403. We will do more work in this aspect.

*p. 17, l. 400: What is meant with "retrieval uncertainty" and how do you estimate it?*

The retrieval uncertainty is created by different retrieval algorithms associated with the retrieved quantities, and we indicate in lines 420-421.

*p. 18, l. 414-415: Please add a sentence explaining what an error of 1 dB corresponds to.*

This sentence has been added, as shown in line 435.

*p. 18-19, l. 414-438: Your error unit seems to be wrong. For example, according to Figure 10 (top left) for true IWC of 10^-5 g/m^3 the maximum error ratio is about 10^-4, which corresponds to -4 B or -40 dB. Your error values in dB are off by a factor of 10.*

I agree that the unit of logarithmic error definition in Eq(14) should be B instead of dB, and we have corrected all the units in both manuscript and figures.

*p. 18-19, l. 430-438: Please discuss the CDFs of Figure 13 or remove them.*

The CDF plots has been deleted.

*p. 19-20, l. 439-463: Except for the IQR the plots of Figure 14 seem to show no added value. Remove them or show the added value.*

The PDF plots are good, but we always need quantitative parameters to assess the retrieval accuracies and make the comparison more straightforward. The top two panels are used to directly compare with the results in Figure 11 in Pfreundschuh et al., (2020), and the bottom panels show the statistics of the absolute retrieval errors, and they serve as the ultimate quantitative parameters for the assessments. We will keep these plots here.

*p. 20, l. 459: Please define explicitly and explain the median fractional bias. An equation could be helpful.*

A sentence saying that "50% of the retrievals have an error less than the median error, and 50% have a larger error" is added, as shown in lines 478-479.

*Section 6 Summary and conclusions*

*Please add some (short) comments comparing your results to Pfreundschuh et al. (2020, AMT) and about your main retrieval assumptions (viewing geometry, footprint sizes, no liquid ...).*

The comments are added in lines 510-512.

*Figures and Tables*

*Figure 1:*
*Please mark in the spectrum plot the relevant spectral lines and features. For example, add an 'O2' to the 118 GHz line and so on.*

The spectral features have been added.

*Figure 2:*
*Within the study, you do not distinguish between ice and snow. Therefore, replace "Frozen (Ice + Snow) Water Content" with 'ice water content'. Furthermore, replace "water content (WC)" with 'ice water content' (IWC).*

The text has been revises.

*Figure 3:*
*Combine ice water path and snow water path.*

Done

*Figure 8:*
*Replace "Ice + Snow Water Content" and "Ice + Snow Number Concentration" with 'Ice water content' and 'Number concentration', respectively.*

Done

*Figure 9:*
*Unit of the y-axis is wrong.*

*Figure 10:*
*IWC and IWP units are wrong.*

*Figure 11:*
*IWC and IWP units are wrong.*

*Figure 12:*
*IWC and IWP units are wrong. Furthermore, the error unit is wrong. The given numerical values correspond to 'B' not 'dB'.*

*Figure 13:*
*The unit for the PDF seems to be missing.*

*Figure 14:*
*The unit of the y-axis is missing.*

The units are all corrected.

---

## Referee Report (RR1)

Comment on amt-2021-2

The authors addressed almost all of my questions and remarks accordingly.

There is one question/remark which still needs an answer and there are two smaller (technical) issues.

Considering my comment on **p. 17, l. 400 (version from August) What is meant with "retrieval uncertainty" and how do you estimate it?**

*Author response:*
*The retrieval uncertainty is created by different retrieval algorithms associated with the retrieved quantities, and we indicate in lines 420-421.*

The authors explained what the retrieval uncertainty but they still did not explained how do they estimate it.

**Smaller issues:**
Figure 3 (p.8): "Integrated water content for ice and snow particles for the selected latitudinal transect and the corresponding brightness temperature…" Caption is inconsitent with figure.

Figure 14 (p. 26): The unit for the PDF (y-axis) still seems to be missing.

---

## Author Response (AR2)

**Responses to Comments**

We sincerely thank Editor Eriksson, and an anonymous reviewer for the constructive and thoughtful comments.

Comment are in *blue italic lettering*, responses in black.

**Editor Comments**

*Sec 2.1: To avoid confusion, clarify already here (or just here) that the simulations and results assume both instruments to be nadir looking and have a very high horizontal resolution (as you don't cover footprint inhomogeneities). As it is now, you present two instruments, but simulate something else.*

Thanks for your suggestion. The assumptions on the active and passive instruments have been added in Sec. 2.1, as shown in lines 84-88. The discussions on this aspect in other paragraphs have been deleted.

*Line 75 + Table 1: The term "desired noise characteristics" is vague. Are these values target, breakthrough or something else more quantitative? Who has set these values? Are they +-1 std dev, or something else?*

The noise characteristics are disclosed by the "AtmOS Microwave Radiometer Instrumentation RFI.PDF" document on the https://aos.gsfc.nasa.gov/gallery.htm website. They show the desired and target radiometric resolution, and the former is used in this study. The noise quantity for the 310GHz is not given, and it is assumed to be 1.5K here. More details are added in the manuscript, as shown in Lines 76-78 and the caption of Table 1.

*Sec 3: Add references to be more clear about what you have taken from the literature and what you have added yourself. I would suggest adding references on at least lines 144, 158 and 188.*

The references have been added, as shown in Lines 143, 148, 157, 162, and 185.

*Fig 3: In panel titles it says "Ghz". Should be "GHz".*

The error in Fig. 3 has been corrected.

*Lines 170-178: Here I want to clarify that applying OEM in ARTS is not restricted to the PSD of De la Noe et al. For example, MGD can be applied in multiple ways. And as far as I see, your prior assumptions could also be applied inside ARTS. More generally, I don't think your argumentation for not doing OEM is fully correct. I would say that what you bring up on lines 179-180 is the critical aspect. That the assumptions behind OEM are not valid for retrieval of hydrometeor properties.*

The statements for the reason have been modified, and I only use the points given in Pfreundschuh et al., 2020, as shown in Lines 175-178.

*Line 344: I don't see why this fact would lead to a stronger constrain for ice inside the altitude region.*

This sentence has been deleted, as shown in the discussions around Lines 340.

*Line 348: Here it becomes clear that the retrieval database contains dry profiles. Maybe I miss something, but do you impose any constrain between the presence of hydrometeors and relative humidity? Or can there be a high IWC where the relative humidity is very low?*

That is a good idea to use the correlations between the IWC and relative humidity. We do not use this information in this study. Since the correlations are right inside the precalculated retrieval database, it is definitely worth investigating. Thanks for giving this hint.

*Line 368: Are the results in Fig 8 consistent with the DOFs obtained in Eriksson et al (2020)?*

I double-checked the program and use the Typhon package to calculate the IWV now. Even though, the discrepancies can still be seen for the dry atmosphere (IWV<42 kg m-2) with low IWC(IWC<100 g m-2), where Fig. 8 shows higher DoFs in this area than the results in Eriksson et al (2020). I think there are two possibilities to explain the disagreement. First, we estimate the DoF using the atmospheric profiles that only contain ice cloud profiles, and liquid hydrometer species are excluded. Second, the random atmosphere/cloud cases in each IWP-IWV bin here are likely to be denser than that in your database since the figure in Eriksson et al (2020) spreads a much broader IWP and IWV range. That implies that more microphysics variabilities may exist in each bin here and therefore, more diversity in the simulated BT is possible. We have added some statements saying that the DoF estimation is likely to be different if we use more complicated atmospheric profiles, as shown in Lines 363-365 and the caption of Fig. 8.

*Line 369: I would suggest calling the section: 5 Results and discussion*

The title of this subsection has been modified, as shown in line 366.

*Figures 11-13 take a lot of space. Are both Fig 11 and 13 really needed? If yes, place them after each other as they are so similar.*

Fig. 13 has been deleted, and the arguments have been combined into the discussion for Fig. 11.

*Fig 12: As Referee #1 pointed out, you don't describe how you define your retrieval uncertainty. Accordingly, what to expect in Fig 12 is not clear. Most importantly, does your retrieval uncertainty include what Rodgers denotes as smoothing error? If no, then I don't see the point in Fig 12 because then you don't know what to expect. You can get very high values in Fig 12 and all is OK (if the total error is dominated by the smoothing term). If yes, then I don't agree with your analysis. Assuming Gaussian errors, there should be a few points above 3 (ie outside of +-3 std dev). Why do you expect that most values should be around 1? On a linear scale, the distribution should peak at 0. Further, I don't think you need all panels here, just some examples should suffice.*

Sorry for not illustrating it clearly. All uncertainties estimations are calculated based on the ensemble approach to finding the standard deviation using the Bayesian MCI. For the radiometer-only retrievals, the last ensemble in the EnPE implementation is used. For the radar-only and synergistic retrievals, the random cases from the Cholesky decomposition are used. The difference is that the random cases have the same weights in the radar-only retrievals, but they are weighted according to the BT disagreements for the combined retrievals. Since the covariance matrix in Eq. (4) is derived by combining the prior Gaussian PDF and the conditional Gaussian PDF, the

smoothing error describing the prior uncertainty should be included. I agree that the analysis of delta error is not appropriate. This figure has been deleted until more deep investigations about how to use the uncertainty as a diagnostic parameter are conducted in future work.

*Line 464: If you think that the selection of the initial state is critical, you have not solved the minimisation problem properly.*

I agree with this point. Thanks for pointing it out. This sentence has been deleted, as shown in Lines 432-437.

*Lines 504-505: It says "a tropical transect" on both lines. The same, or different ones?*

The sentence has been corrected. As shown in Line 475-476.

*There are also a number of language issues, but I expect that those problems are handled in the copy-editing.*

The manuscript has been polished and many grammar mistakes are corrected. Please check it again.

*As a final remark, I can mention that our new article on joint passive and active retrievals https://amt.copernicus.org/preprints/amt-2021-306 got accepted one week ago and should come out in AMT relatively soon. Contact me if you want to see the final manuscript version already now.*

I have read this paper and it is fantastic. Thanks for sharing this work. Studying in Prof. Buehler's group is one of the best experiences I have ever had in my life. Hopefully I could have the opportunity to work with the ARTS group again in the future. All the best wishes for the ARTS community.

**Reviewer 1 Comments**

*Considering my comment on p. 17, l. 400 (version from August) What is meant with "retrieval uncertainty" and how do you estimate it?*

*Author response:*

*The retrieval uncertainty is created by different retrieval algorithms associated with the retrieved quantities, and we indicate in lines 420-421.*

*The authors explained what the retrieval uncertainty but they still did not explained how do they estimate it.*

Sorry for the poor illustrations of the retrieval uncertainties. More explanations are given in the response to the editor's comments above. Please check it.

*Figure 3 (p.8): "Integrated water content for ice and snow particles for the selected latitudinal transect and the corresponding brightness temperature..." Caption is inconsitent with figure.*

The caption of Fig. 3 has been corrected.

*Figure 14 (p. 26): The unit for the PDF (y-axis) still seems to be missing.*

The unit for the PDF has been added.